# Engineering Nanopatterned Structures to Orchestrate Macrophage Phenotype by Cell Shape

**DOI:** 10.3390/jfb13010031

**Published:** 2022-03-14

**Authors:** Kai Li, Lin Lv, Dandan Shao, Youtao Xie, Yunzhen Cao, Xuebin Zheng

**Affiliations:** 1Key Laboratory of Inorganic Coating Materials CAS, Shanghai Institute of Ceramics, Chinese Academy of Sciences, 1295 Dingxi Road, Shanghai 200050, China; likai@mail.sic.ac.cn (K.L.); lvlin@mail.sic.ac.cn (L.L.); shaodandan@student.sic.ac.cn (D.S.); xieyoutao@mail.sic.ac.cn (Y.X.); yzhcao@mail.sic.ac.cn (Y.C.); 2Center of Materials Science and Optoelectronics Engineering, University of Chinese Academy of Sciences, 19 Yuquan Road, Beijing 100049, China

**Keywords:** nanopillar, nanopit, macrophage cell shape, phenotype, integrin β1

## Abstract

Physical features on the biomaterial surface are known to affect macrophage cell shape and phenotype, providing opportunities for the design of novel “immune-instructive” topographies to modulate foreign body response. The work presented here employed nanopatterned polydimethylsiloxane substrates with well-characterized nanopillars and nanopits to assess RAW264.7 macrophage response to feature size. Macrophages responded to the small nanopillars (SNPLs) substrates (450 nm in diameter with average 300 nm edge-edge spacing), resulting in larger and well-spread cell morphology. Increasing interpillar distance to 800 nm in the large nanopillars (LNPLs) led to macrophages exhibiting morphologies similar to being cultured on the flat control. Macrophages responded to the nanopits (NPTs with 150 nm deep and average 800 nm edge-edge spacing) by a significant increase in cell elongation. Elongation and well-spread cell shape led to expression of anti-inflammatory/pro-healing (M2) phenotypic markers and downregulated expression of inflammatory cytokines. SNPLs and NPTs with high availability of integrin binding region of fibronectin facilitated integrin β1 expression and thus stored focal adhesion formation. Increased integrin β1 expression in macrophages on the SNPLs and NTPs was required for activation of the PI3K/Akt pathway, which promoted macrophage cell spreading and negatively regulated NF-κB activation as evidenced by similar globular cell shape and higher level of NF-κB expression after PI3K blockade. These observations suggested that alterations in macrophage cell shape from surface nanotopographies may provide vital cues to orchestrate macrophage phenotype.

## 1. Introduction

Advances in nanotechnology are increasing the likelihood of programmable nano-bio interfaces, particularly in the novel nanotopographical design to foster cellular modulation, genome editing and biomolecular delivery [1,2]. Engineered nano-bio interfaces fabricated by novel nanopatterning technology have improved the fundamental understanding of cellular functions at the nanoscale [3,4,5]. Nanopatterned structures are now providing great advantages in manipulating cellular functions and processes in a more controlled manner, which assists the translation into practical applications [6].

In the field of tissue engineering, three basic nanopatterned geometries, nanogrooves, nanopillars and nanopits, are frequently used as a means to investigate stem cell attachment and to control cell differentiation and phenotype in vitro [7]. A study by Yim et al. reported that human mesenchymal stem cells (hMSCs) grown on nanogrooves were preferentially differentiated into a neural lineage as evidenced by the existence of nestin and synaptophysin markers and by the regulation of MAP2 [8]. Dalby et al. reported that hMSCs grown on slightly irregular nanopits produced considerable quantities of bone-specific extracellular matrix proteins osteopontin and osteocalcin. In contrast, cells grown on completely ordered or random substrates did not result in the expression of those two proteins [9]. Fu et al. reported that enhanced osteogenic differentiation of hMSCs was observed on patterned nanopillars with shorter heights (0.97 um) [10].

Although tissue engineering strategies highly focus on stem cells, immune cells are equally essential for tissue repair and regeneration [11]. Of them, macrophages are early responders to tissue injury and are extremely plastic cells involved in host defense, immune regulation and tissue repair [12]. To achieve such diverse functions, macrophages polarize to different phenotypes, ranging from M1 to M2 (pro-inflammatory to anti-inflammatory/pro-healing, respectively), in response to various microenvironment stimuli [13]. The effect of physical stimuli, such as mechanical stiffness and topography, on macrophage behaviors has long been known [14]. Various types of nanotopographies have been shown to have an effect on macrophage cell shape and phenotype, especially when the topography is highly ordered [5]. In a study by Frances et al., elongated cell morphology alone was found to induce bone marrow-derived macrophages (BMMs) polarization into the anti-inflammatory M2 phenotype with increased expression of arginase-I (ARG-I) and mannose receptor CD206 [15]. A surface with nanopatterned grooves drove BMMs polarization to M2 phenotype, and the level of anti-inflammatory cytokine interleukin-10 (IL-10) production in macrophages peaked on the substrate with groove widths of 400–500 nm.

While correlations between nanogrooves, macrophage cell shape and phenotype have been established, only few studies used well-defined nanopillars or nanopits for the evaluation of the specific effect of surface parameters (e.g., diameter, height and spacing) in macrophage polarization. A study by Padmanabhan et al. suggested that patterned nanopillars of diameter larger than 200 nm were required to induce morphological changes in BMMs, leading to more elongated and larger cell morphology [4]. Conflicting results have demonstrated the decrease in macrophage cell spreading on nanopillars [16,17]. The overall effect of these nanotopographies on macrophage cell shape and phenotype is unclear. It has however been shown that nanopit and nanopillar substrates could be employed to investigate the role of filopodia formation, focal adhesion dynamics and cytoskeletal organization in a controlled manner [18]. A systematic examination of nanotopography-cell interaction between nanopatterned pillars or pits and macrophages may provide data on cell shape modulation as the underlying mechanisms driving macrophage polarization status changes.

Here, nanopatterned pillars and pits in varying sizes were fabricated on polydimethylsiloxane (PDMS) substrates by replica molding. The effects of nanopattern features on macrophage morphology, phenotype and gene expression profile were evaluated. In order to study the biochemical mechanism underlying nanopattern-cell interactions, nanopattern-induced changes in fibronectin adsorption, integrin expression, focal adhesion formation and expression of downstream effector kinases were analyzed.

## 2. Materials and Methods

### 2.1. Fabrication and Characterization of the Nanopatterned Surfaces

Silicon wafers with highly ordered nanopits (390 and 780 nm pit diameter) in a hexagonal conformation pattern (In-situ Technology Co., Ltd., Shanghai, China) were used as templates. Nanopatterned PDMS substrates were prepared via mixing the curing agent (Sylgard 184, Dow Corning, Midland, MI, USA) with silicone elastomer in an 1:10 ratio by weight, pouring the prepolymer over the patterned silicon wafers (10 mm × 10 mm, 20 mm × 20 mm) and curing 2 h at room temperature and 3 h at 60 °C. Once cured, the PDMS was peeled from the wafer. By replica molding, the large nanopillars (LNPLs) and small nanopillars (SNPLs) substrates were prepared. The SNPLs substrate was further used to fabricate the patterned nanopit (NPT) substrate. A flat PDMS substrate served as control. The surfaces of nanopatterned substrates were observed by field emission scanning electron microscope (FE-SEM, Magellan 400, FEI Company, Hillsboro, OR, USA). Surface topography was analyzed by atomic force microscope (AFM, Bruker Dimension Icon, Karslruhe, Germany). The imaging was conducted in tapping mode with a cantilever of spring constant of 2.8 N m^−1^ with a resonance frequency of 75 kHz. The images were taken at a scan size of 3 μm × 3 μm. Surface wettability was measured by contact angle meter (SL200B, KINO Industry Co., Ltd., Boston, MA, USA).

### 2.2. Effects of the Nanopatterned Structures on Fibronectin Distribution and Conformation

Each sample was immersed in 20 μg/mL of fibronectin (FN, Sigma, St. Louis, MO, USA) solution for 10 min at 37 °C. After this, each sample was rinsed by phosphate-buffered saline (PBS) to remove non-adsorbed FN. The distribution of adsorbed FN on the surfaces was observed by AFM in a tapping mode. The amount of adsorbed FN was quantitatively measured using the Micro BCA Protein Assay Kit (Beyotime Biotechnology, Nantong, China). RGD availability of FN on the surfaces was assessed by examining the binding of the anti-FN antibody (HFN7.1) with enzyme-like immunosorbent assay (ELISA) as described elsewhere [14]. Considering the difference among the samples in the amount of adsorbed FN, RGD availability of FN was calculated by normalizing the intensity of HFN7.1 to the amount of adsorbed FN.

### 2.3. Cell Culture

The murine-derived macrophage RAW264.7 cell line was obtained from the Cell Bank of Type Culture Collection of Chinese Academy of Sciences (Shanghai, China). The cells were incubated in α-minimum essential medium (α-MEM, HyClone, Logan, UT, USA) supplemented with 1% (*v*/*v*) penicillin/streptomycin (Wisent, Nanjing, China) and 10% (*v*/*v*) fetal bovine serum (FBS, Wisent) in an incubator at 37 °C with 5% CO_2_. The medium was refreshed every two days, and the cells were passaged upon reaching 80% confluence.

### 2.4. Effects of the Nanopatterned Structures on Macrophage Cell Shape and Phenotype

#### 2.4.1. Cell Morphology

RAW264.7 cells were cultured on the samples in 48-well plates at a density of 2 × 10^4^ cells/well. After incubation for 24 h, the cells were fixed overnight in 2% (*v*/*v*) glutaraldehyde, dehydrated with a graded ethanol series and air-dried. FE-SEM was used for cell morphology observation. Average cell area of cells on each sample was calculated using 50 measurements from three random areas of a SEM image.

#### 2.4.2. Cytoskeleton and Focal Adhesion Observation

RAW264.7 cells were seeded onto each sample in 48-well plates at a density of 2 × 10^4^ cells/well. After incubated for 3 days, the cells were treated by 4% (*v*/*v*) paraformaldehyde fixation and 0.5% (*v*/*v*) Triton X-100 permeabilization. After incubation in blocking buffer (5% bovine serum albumin in 0.1 M PBS), the cells were treated by primary antibody against vinculin (1:150, Cell Signaling Technology, Danvers, MA, USA) diluted in blocking buffer overnight at 4 °C. Afterward, the cells were incubated with fluorescence-conjugated secondary antibody against vinculin (1:250, Cell Signaling Technology) for 1 h. Meanwhile, the cells were treated by DAPI (Molecular Probes) and rhodamine phalloidin (Molecular Probes) for nucleus and actin staining, respectively. Confocal microscope system (TCS SP5 II, Leica, Nussloch, Germany) was used for immunofluorescence observation. Elongation factor was calculated as the length of the cell’s long axis divided by the length of the short axis across the nucleus as described by McWhorter et al. [15]. Approximately one hundred cells were evaluated for the quantitative analysis of cell morphology changes.

#### 2.4.3. Flow Cytometry

RAW264.7 cells were cultured on the samples in 6-well plates at a density of 5 × 10^5^ cells/well and collected after 3 days. After centrifugation and re-suspension, the cells were stained with anti-mouse CD11c-FITC and anti-mouse CD206-PE (Thermo Scientific, Waltham, MA, USA) at 4 °C in darkness for 30 min. The flow cytometry was performed using a BD FACSAria flow cytometry system (BD Biosciences, Franklin Lakes, NJ, USA). FlowJo software was employed for data analysis.

#### 2.4.4. Quantitative Real-Time Polymerase Chain Reaction (qRT-PCR) Assay

RAW264.7 cells were incubated on the samples in 6-well plates at a density of 5 × 10^5^ cells/well for 3 days. Total RNA was isolated using the Trizol reagent (Life Technologies, Carlsbad, CA, USA) and reverse-transcribed by a reverse transcription kit according to the manufacturer’s instructions (TaKaRa Biotechnology, Beijing, China). qRT-PCR was performed using Applied Biosystems 7500 RT-PCR system (Applied Biosystems, Waltham, MA, USA) with a PCR kit (Applied Biosystems) to measure the expression of the integrin receptors (*Itga5*, *Itgb1*), the surface markers (*Cd11c*, *Ccr7*, *Cd206*, *Cd163*), the pro-inflammatory cytokines (*Tnfa*, *Il6*, *Il1b*) and the anti-inflammatory cytokines (*Il10*, *Il1ra*). All mRNA values were normalized against the housekeeping gene (GAPDH). All primer sequences are presented in Table 1.

### 2.5. Western Blot

RAW264.7 cells were incubated on the samples in 6-well plates at a density of 5 × 10^5^ cells/well for 3 days. The cells were collected and lysed with Ripa buffer (Beyotime Biotechnology, China), which was supplemented with 1% (*v*/*v*) protease inhibitor. After denaturation and separation, the proteins were treated by primary rabbit Anti-phospho-PI3-kinase p85-α/γ (pTyr467/199, Sigma), phospho-Akt Ser473 (D9E, CST), NF-κB p65 (D14E12, CST) and GAPDH (CST), respectively. Afterward, anti-rabbit IgG secondary antibodies conjugated to horseradish peroxidase (Proteintech Group, Inc., Rosemont, IL, USA) were used. An ECL system (Amersham Pharmacia Biotech, Piscataway, NJ, USA) was employed to reveal the protein bands.

Small interfering RNA (siRNA) was utilized to inhibit integrin β1 expression. RAW264.7 cells were incubated on the samples in 6-well plates at a density of 5 × 10^5^ cells/well and then infected with siRNA (GenePharma, Shanghai, China) to specifically knockdown β1. Inhibition of PI3K was performed by adding PI3K inhibitor LY294002 (L9908, Sigma) at a concentration of 10 and 20 μM. RAW264.7 cells in the absence of the inhibitor served as control. After inhibition, western blot assay of p-PI3K, p-Akt and NF-κB was performed as described above.

### 2.6. Statistical Analysis

The data were shown as mean ± standard deviation (SD) of three independent experiments (sample size n = 3 performed in triplicate). One-way ANOVA followed by SNK post hoc test was employed for analysis of statistical difference. Significance was indicated by * (*p* value below 0.05).

## 3. Results and Discussion

### 3.1. Surface Characterization

PDMS substrates with different surface topographies were fabricated by a replica molding method with three different patterns. Generally, the topographical features as observed by SEM and AFM (Figure 1) consisted of a hexagonal placement of cylindrical nanopillars or nanopits. The large nanopillars (LNPLs) were 400 nm high pillars with a diameter of 700 nm and an average edge-edge distance of 800 nm; the small nanopillars (SNPLs) were 150 nm in height and 450 nm in diameter with average 300 nm edge-edge spacing; the nanopits (NPTs) were 700-nm-diameter pits (150 nm deep, average 800 nm edge-edge spacing). A flat PDMS substrate served as control. The water contact angle of the flat, LNPLs, SNPLs and NPTs substrates was measured as 118.28 ± 0.22°, 120.29 ± 1.94°, 112.71 ± 1.48° and 135.33 ± 2.30°, respectively.

### 3.2. Effects of Nanopatterned Topographies on Macrophage Morphology

To monitor changes in cell morphology caused by nanopatterned topographies, RAW 264.7 macrophage cells were cultured on nanopatterned or flat PDMS substrates for 24 h. In general, cells exhibited heterogenous morphologies; nevertheless, certain morphologies were more predominant on specific nanopatterned substrates, which were presented in SEM images (Figure 2).

Macrophages on the flat control displayed a native globular shape with almost no protrusions. For LNPLs, the morphological responses of macrophages due to the topographical features were slightly but not considerably influenced by the nanopillars, and thus were close to those found on the flat control. A few cells with an elongated shape were observed as well, which had long and thin extensions following the nanopillar array. Such extensions (i.e., confined by the spacing between the nanopillars) were not found on the flat surface. The magnified image (8000 times) for LNPLs showed that the cell body was mostly suspended on top of the nanopillars, whereas the cell gripped the nanopillars with filopodia to induce some pillar bending. On SNPL substrate, most cells displayed a well-spread polygonal shape with the formation of large lamellipodium. Moreover, filopodia can be seen emerging from the lamellipodium and locate the end to the nanopillars, as shown in the 8000 times SEM image. In contrast, cells on NPT substrate were more frequently spindle-shaped. The filopodial formation was observed between the nanopit features at the inter-pit region in the magnified image (8000 times).

Fluorescence microscopy images (Figure 3A) were in agreement with SEM observations of macrophage adaptation to the different nanotopographies. In addition, the absence of organized actin stress fibers was observed for the macrophages, which was consistent with previously reported observations. Instead, the cells exhibited diffuse actin staining with some clustering, particularly around their periphery. A quantitative analysis of cell morphology changes (Figure 3B) was performed by calculating the projected spreading area of the RAW264.7 cells on each sample and their elongation factor. The spreading cell area remained constant for the macrophage grown on the flat control, LNPL and NPT substrates. SNPLs promoted a statistically significant increase in spreading cell area. Concerning the elongation factor, the value is high when cell shape is elongated and decreases when the cell becomes more polygonal. Cells on the NPTs substrate showed a considerably higher degree of elongation compared with cells on the other three substrates. Macrophages on the SNPLs had the lowest degree of elongation.

The dimensions of the nanopillars, such as diameter, height and spacing, were shown to influence the cell’s morphology and spreading. When these dimensions are such that cell membrane cannot accommodate surface recess, the cell is localized on the top of the nanopillars. Generally, this is achieved when the nanopillars are densely packed and their height is more than 40 nm. These observations were in agreement with the SEM results in our study. When the height and density of the nanopillars are sufficient to isolate the cells from the underlying planar substrate, the effect of the nanopillar edge-edge spacing and diameter becomes the determining factors on integrin clustering regulation and subsequent focal adhesion formation [19]. To facilitate integrin activation and clustering in the cells suspended on a nanopillar array, the feature diameter must exceed 70 nm, which was verified by multiple works using nanopillar arrays >400 nm in height [20]. In this work, considering the feature diameter of LNPLs and SNPLs both exceeding 70 nm, it is reasonably to believe that edge-edge spacing plays a vital role in determining cellular adhesion morphology. Compared to SNPLs with edge-edge distance of 300 nm, increasing the distance to 800 nm in LNPLs markedly reduced macrophage cellular adhesion. This indicated that a higher interpillar distance might inhibit integrin clustering and focal adhesion formation at the bridging site between two adjacent nanopillars.

Similarly, pitted topographies have been observed to exert different effects on cell adhesion in vitro, depending on pit depth, spacing and diameter. Increasing pit diameter over 70 nm has been shown to perturb integrin clustering [21,22]. This was consistent with the results of cellular adhesion on nanpillar arrays with feature diameter of <70 nm. When pit depth exceeds 100 nm, the sites of focal adhesions seem to be confined to the inter-pit area [23,24]. This might explain the spindle-shaped macrophage cell morphology on NPTs and the filopodial formation observed at the inter-pit region.

### 3.3. Correlation of Cell Shape and Macrophage Polarization Status

To investigate whether cell shape may have a direct effect on macrophage polarization status, we employed flow cytometry to calculate the percentage of M1 and M2 cells on different substrates by measuring the level of the surface markers CD11c and CD206, respectively. As shown in Figure 4A, a decreased percentage of M1-type macrophages expressing CD11c was found for the SNPLs (44.6%) and NPTs (49.8%) compared with the flat control (66.2%) and LNPLs (61.0%). In addition, the percentage of CD206-positive cells increased from 31.2% for the flat control to 37.8%, 60.9% and 65.0% for the LNPLs, SNPLs and NPTs, respectively.

To confirm the phenotypic polarization of RAW264.7 macrophages, we measured the gene expression level of M1/M2 phenotypic markers by qRT-PCR (Figure 4B). There was no significant difference between the flat control and LNPL substrates in the expression of *Ccr7* and *Cd11c* (M1 surface markers) as well as *Tnfa*, *Il6* and *Il1b* (pro-inflammatory genes). However, the cells on the LNPLs led to significant up-regulation of *Cd163* and *Cd206* (M2 surface markers) as well as *Il10* and *Il1ra* (anti-inflammatory genes) expression when compared with the flat control. Cells grown on the SNPLs and NPTs showed further increases in *Cd206* and *Il1ra* expression. Compared to the flat control and LNPLs, considerably decreased expression of *Ccr7*, *Il6* and *Il1b* were observed for the SNPLs and NPTs. SNPLs and NPTs exerted a greater effect on the macrophage polarization status through inducing the M2 phenotype. Together, these data suggested that macrophage cell shape was associated with the macrophage polarization status; besides, well-spread cell shape induced by SNPLs and elongated shape induced by NPTs correlated with M2 polarization.

Growing evidence has revealed that morphology is heavily associated with phenotype in certain macrophages; however, the phenomenon may not be universal. We and others have observed that while M2-polarized RAW264.7 exhibited well-spread cell shape, other macrophages did not display such dramatic changes [25,26,27]. Likely contributing to conflicting reports were several factors, including the differences between murine and human macrophages used in different reports, and the sensitivity of macrophages to different substrates including polymers, ceramics and metals [28,29,30,31].

### 3.4. Biochemical Cues Underlying Cell-Nanopattern Interactions Govern Macrophage Polarization Status

The initial adsorption of extracellular matrix (ECM) proteins, such as FN, which forms a temporal matrix on biomaterial surface, becomes a vital link between the biomaterial and host response. Therefore, it is imperative to know the manner by which the adsorbed proteins interact with the nanopatterned surface when determining the subsequent behavior of macrophages. The distribution of adsorbed FN was investigated by AFM in a tapping mode (Figure 5A). Globular aggregates appeared on the flat surface. In contrast, interconnected nanofibrils were observed on the nanopillar array surfaces. Imbedded FN molecules were found in the nanopits. The adsorbed FN proteins have been reported to have different configurations, such as filamentous or globular and extended or close-packed forms, depending on the local environment and the substrate [32]. Thus, the native conformation of globular FN in solution was maintained following adsorption onto the flat control, while FN proteins were adsorbed in an extended conformation on the nanopillar array surfaces. The distinct distribution of FN at the material interface might have consequences at the molecular level for availability of the integrin binding domain of FN (FNIII_9-10_) [33].

The RGD amino acid sequence, located in the FN repeat III_9-10_, is of great importance in modulating cell adhesion, enhancing α5β1 integrin binding to FN [34]. The RGD availability of FN on the sample surfaces was evaluated by examining the binding of the anti-FN antibody (HFN7.1). As shown in Figure 5B, the intensity of HFN7.1 for the nanopatterned substrates was greatly increased compared to the flat control. Moreover, the intensity was increased on SNPLs in comparison to LNPLs and NPTs. However, the difference did not reach statistical significance. Generally, α5β1 integrin is considered to be a major receptor for macrophage adhesion to FN [35]. Figure 5C showed that the expression level of integrin β1 (*Itgb1*) was significantly higher on the nanopatterned substrates compared to the flat control. This was consistent with the result of the RGD availability of FN. However, among the nanopatterned substrates, the SNPLs and NPTs showed higher expression of *Itgb1* than the LNPLs. Subsequent to binding to ECM proteins, integrins cluster and develop into complexes called focal adhesion. Feature dimension on the nanopatterned substrates might play a role in integrin clustering and result in different *Itgb1* expression. Massia and Hubbell found that clustering of at least 6 RGD-ligand-occupied integrin heterodimers per micron (corresponding to a spacing of 440 nm) was sufficient to support cell spreading [36]. Previous work with defined arrays of bound RGD fragments demonstrated that substratum-integrin interactions were disrupted when the integrin spacing was in the range 70–300 nm [3,37,38]. Hence, it can be inferred that LNPLs with nanopillar spacing of 800 nm disrupted the lateral spacing of integrin clustering, which restricted focal adhesion formation between two adjacent nanopillars and the macrophages had to extend their filopodia to find more adsorbed proteins. By contrast, SNPLs with nanopillar spacing of 300 nm and NPTs with inter-pit region of 800 nm could facilitate integrin clustering, thus storing focal adhesion formation.

Stable focal adhesion requires integrin clustering and stability of the binding between the two (integrin cluster size), and it also involves the stabilization of the cytoplasmic side with reinforcement of adhesion proteins, including vinculin, paxillin, talin and focal adhesion kinase [39]. In this work, we employed vinculin as a marker of focal adhesion considering that it is recruited at the adhesion site via integrin β1 receptor. The localization of vinculin was probed via immune-fluorescence microscopy (Figure 5D). Relatively smaller focal adhesions were observed on the flat control surface. For LNPLs, focal adhesions were localized regularly along the cell periphery, while cells on SNPLs formed adhesion contact not only at cell periphery but also within the lamellipodium leading edge. Focal adhesion on NPTs were distributed predominantly to the cell periphery; meanwhile, they were observed to be involved in cell protrusions.

NF-κB plays a vital role in the innate immune response [40,41]. As shown in Figure 6A, enhanced levels of NF-κB protein were measured on the LNPLs and flat substrates when compared with the SNPLs and NPTs. This was consistent with the results of macrophage polarization status on the substrates. NF-κB subunits have sites for phosphorylation and other post-translational modification which are vital for crosstalk with other signaling pathways. The crosstalk can occur through various kinases, including glycogen synthase kinase and Akt triggered by phosphoinositide 3-kinase (PI3K), which regulate NF-κB transcriptional activity [42]. In this work, the phosphorylation levels of PI3K and Akt on the SNPLs and NPTs substrates were higher than those of the LNPLs and flat substrates. Similar results were observed by Zhao et al. that PI3K/Akt pathway negatively regulated NF-κB pathway and the expression of inflammatory genes [43]. As shown in Figure 6B, the phosphorylation of PI3K/Akt was inhibited after β1-specific knockdown, which indicated that β1 integrin was implicated in PI3K/Akt pathway activation.

To further investigate the effect of PI3K/Akt pathway on the transactivation potential of NF-κB, NF-κB transactivation was monitored in the presence of PI3K inhibitor LY294002. As shown in Figure 6C, blockade of PI3K increased NF-κB activation. Moreover, higher concentration of LY294002 resulted in higher level of NF-κB activation. These confirmed that PI3K/Akt pathway negatively regulated NF-κB activation. However, it is necessary to point out that the effect of PI3K/Akt pathway on NF-κB activity often highly relies on the micro-environment or the cell type and that even an opposite effect can occur in distinct cell types. For instance, the effect of Akt on NF-κB was reported, which was activating in epithelial cells, but can be inhibitory in macrophages [44,45]. PI3K has also been recognized to regulate actin assembly and cytoskeleton organization facilitating cell spreading [27,46,47]. After PI3K inhibition, the macrophages on all the substrates showed globular morphology (Figure 6D). This confirmed the pivotal role of PI3K in promoting cell spreading and thus explained the more spreading cell morphology on SNPL and NPT substrates.

Because macrophages bridge innate immunity and adaptive immunity, a new paradigm was established to harness macrophage responses by immunomodulating biomaterials for endogenous repair. During macrophage-material interaction in vivo, the material surface acts as a temporary niche for macrophages to reside in, and the properties of the material surface weigh in on macrophage phenotypes. A broad range of material surface properties, such as pore size, shape and geometry, stiffness, topography and surface modification (e.g., hydrophilicity, integrin engagement), have been proven to modulate macrophage behavior and tune implantation outcomes. Although a lot of these studies observed correlations between material designs and macrophage phenotypes, a refocus on directing macrophage behavior to achieve suitable immune responses needs to be emphasized.

## 4. Conclusions

Utilizing a nanopatterning approach to modulate macrophage cell shape directly, we demonstrated here that elongation and well-spread cell shape led to expression of M2 phenotypic markers and downregulated expression of inflammatory cytokines. Changes in macrophage phenotype polarization from surface nanotopographies depended on a mechanism involving changes in cell adhesion and shape. In contrast to SNPLs, a higher interpillar distance in LNPLs (800 nm) inhibited integrin clustering, *Itgb1* expression, and focal adhesion formation, resulting in reduced macrophage cellular adhesion. The NPTs substrates with 150 nm pit depth confined the sties of focal adhesions to the inter-pit area (average 800 nm edge-edge spacing), resulting in elongation cell morphology. SNPLs with smaller interpillar distance and NTPs with higher inter-pit area facilitated the exposure of availability of integrin binding region of FN, expression of *Itgb1*, and formation of focal adhesion. Higher expression of *Itgb1* in macrophages on the SNPLs and NTPs activated the intracellular PI3K/Akt pathway, which promoted macrophage cell spreading and negatively regulated NF-κB activation as evidenced by similar globular cell shape and higher level of NF-κB activation after PI3K blockade. Our findings highlighted the potential of using nanopatterned polymers to modulate macrophage functions and thus the foreign body response.

## Figures and Tables

**Figure 1 jfb-13-00031-f001:**
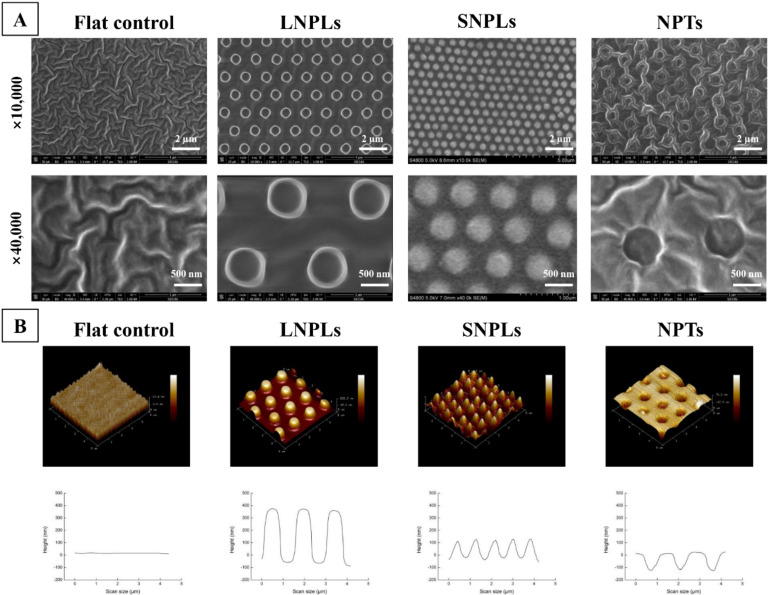
Surface characterization of the nanopatterned and flat substrates. (**A**) SEM images of the substrate surfaces. (**B**) AFM images of the substrate surfaces: three dimensional images (upper row) and height profiles (bottom row).

**Figure 2 jfb-13-00031-f002:**
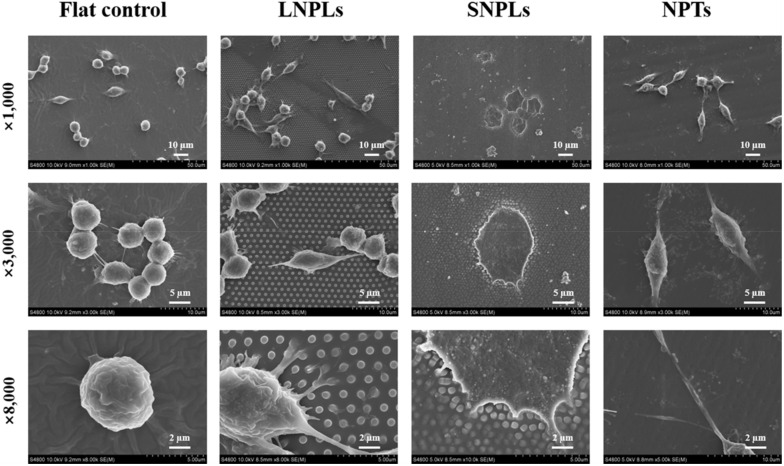
SEM images of RAW264.7 cells on the nanopatterned and flat substrates after 1 day of incubation.

**Figure 3 jfb-13-00031-f003:**
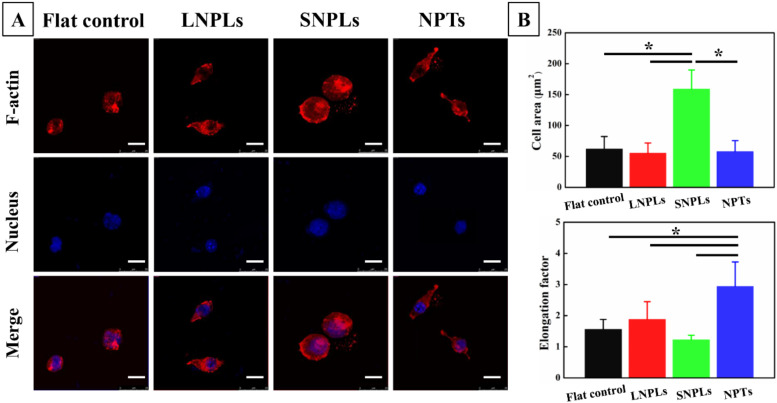
(**A**) Typical immunofluorescence images of RAW264.7 cells on the nanopatterned and flat substrates after 3 days of incubation. Scale bar = 15 μm. (**B**) Quantitative analysis of cell morphology changes. Asterisks indicate significant difference (* *p* < 0.05).

**Figure 4 jfb-13-00031-f004:**
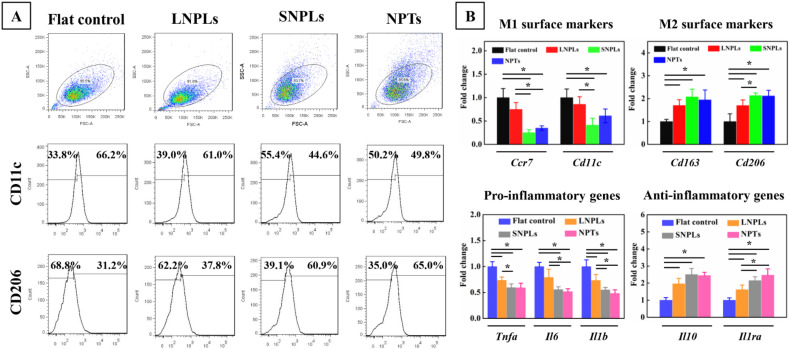
Assessment of macrophage polarization status on the nanopatterned and flat surfaces according to (**A**) flow cytometry assay of M1/M2 surface markers (CD11c, CD206) and (**B**) fold change of gene expression of M1/M2 surface markers (*Ccr7* and *Cd11c* for M1, *Cd163* and *Cd206* for M2), pro-inflammatory cytokines (*Tnfa, Il6*, *Il1b*) and anti-inflammatory cytokines (*Il10*, *Il1ra*). Asterisks indicate significant difference (* *p* < 0.05).

**Figure 5 jfb-13-00031-f005:**
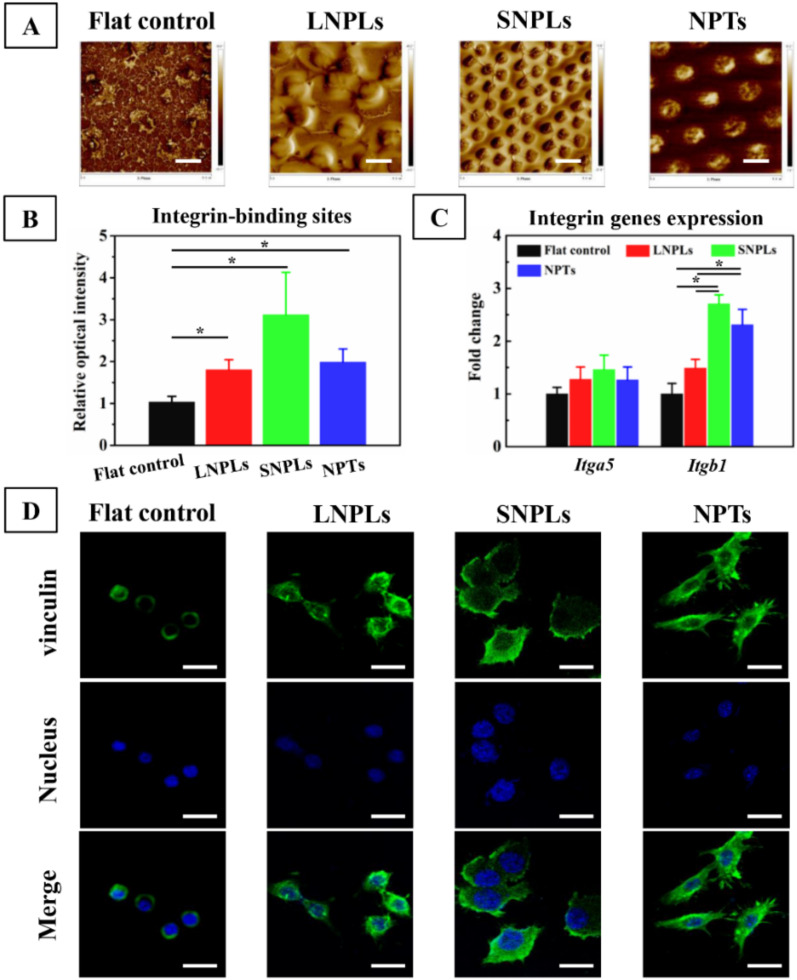
(**A**) FN distribution on the nanopatterned and flat surfaces as observed by AFM. Scale bar = 5 μm. (**B**) Relative intensity of integrin-binding sites of adsorbed FN on the nanopatterned and flat surfaces measured by ELISA. (**C**) Fold change of gene expression of integrin receptors α5 *(Itga5*) and β1 (*Itgb1*)) by RAW264.7 cells on day 3. (**D**) Vinculin (green), cell nucleus (blue) and merged staining images of RAW264.7 cells on the nanopatterned and flat substrates after 3 days of incubation. Scale bar = 20 μm. Asterisks indicate significant difference (* *p* < 0.05).

**Figure 6 jfb-13-00031-f006:**
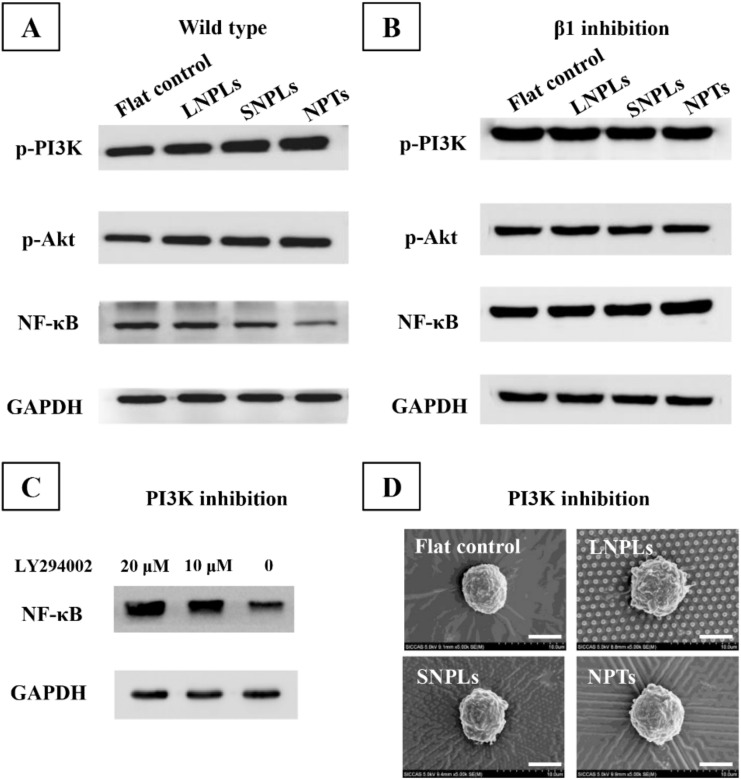
Western blot analysis of p-PI3K, p-Akt and NF-κB in (**A**) wild-type and (**B**) β1 knockdown RAW264.7 macrophages. (**C**) Western blot analysis of NF-κB activation in the presence of PI3K inhibitor LY294002. (**D**) SEM images of RAW264.7 cells after 1 day of incubation in the presence of 20 μM LY294002.

**Table 1 jfb-13-00031-t001:** Sequences of all primers used in this study.

Target Gene	Direction	5′-3′ Primer Sequence
*Itga5*	F	5′-CTTCTCCGTGGAGTTTTACCG-3′
	R	5′-GCTGTCAAATTGAATGGTGGTG-3′
*Itgb1*	F	5′-CGTGGTTGCCGGAATTGTTC-3′
	R	5′-ACCAGCTTTACGTCCATAGTTTG-3′
*Cd11c*	F	5’ -CTGGATAGCCTTTCTTCTGCTG- 3’
	R	5’ -GCACACTGTGTCCGAACTCA- 3’
*Ccr7*	F	5′-TGTACGAGTCGGTGTGCTTC-3′
	R	5′-GGTAGGTATCCGTCATGGTCTTG-3′
*Cd163*	F	5′-ATGGGTGGACACAGAATGGTT-3′
	R	5′-CAGGAGCGTTAGTGACAGCAG-3′
*Cd206*	F	5′-CTCTGTTCAGCTATTGGACGC-3′
	R	5′-CGGAATTTCTGGGATTCAGCTTC-3′
*Il6*	F	5′-ACTCACCTCTTCAGAACGAATTG-3′
	R	5′-CCATCTTTGGAAGGTTCAGGTTG-3′
*Tnfa*	F	5′-CCTCTCTCTAATCAGCCCTCTG-3′
	R	5′-GAGGACCTGGGAGTAGATGAG-3′
*Il1b*	F	5′-TGGAGAGTGTGGATCCCAAG-3′
	R	5′-GGTGCTGATGTACCAGTTGG-3′
*Il10*	F	5′-GACTTTAAGGGTTACCTGGGTTG-3′
	R	5′-TCACATGCGCCTTGATGTCTG-3′
*Il1ra*	F	5′-CATTGAGCCTCATGCTCTGTT-3′
	R	5′-CGCTGTCTGAGCGGATGAA-3′
GAPDH	F	5′-TGACCACAGTCCATGCCATC-3′
	R	5′-GACGGACACATTGGGGGTAG-3′

## Data Availability

Not applicable.

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
