# Peer review of "Engineering Nanopatterned Structures to Orchestrate Macrophage Phenotype by Cell Shape"

_jfb, 2022, doi:10.3390/jfb13010031_

Round 1

Reviewer 1 Report

The submitted study characterizes the reaction of RAW264.7 macrophages to culture on nanopillars and nanopits coated with fibronectin. The manuscript is clearly written and the results support the conclusions made by the authors.

Comments

Materials and methods:

-Description of AFM experiments should give more details e.g. cantilever

-How many cells were evaluated for the quantitative analysis of cell surface shown in Figure 3?

Results:

It would have been good to compare the extent of polarization with the conventional method (induction by cytokines, LPS)

Is the polarization transient or permanent? Can results (not the entire data set) after several days of culture on these surfaces be shown to answer this question.

General:

It is surprising that the study is submitted to the section “Bone Biomaterials”

Reviewer 2 Report

It's a good job, well presented.

my suggestions are:

1. if the work was approved by a research committee? can you put the number in the writing?
2. I suggest dividing the results of the discussion into two sections. Thank you
3. Statistics need to be added at the foot of the figures, for example in figures 3 and 4.
4. Were the Western blot analysis studies (figure 6) performed in triplicate? do you have pictures to show? Western blot quantification was performed? Do you have the results to put in a table or in a figure?
5. I suggest expanding the discussion to highlight the importance of the study

Round 2

Reviewer 1 Report

The authors addressed my comments.

Reviewer 2 Report

no